# Extreme Weather Events Enhance DOC Consumption in a Subtropical Freshwater Ecosystem: A Multiple-Typhoon Analysis

**DOI:** 10.3390/microorganisms9061199

**Published:** 2021-06-01

**Authors:** Chao-Chen Lai, Chia-Ying Ko, Eleanor Austria, Fuh-Kwo Shiah

**Affiliations:** 1Research Center for Environmental Changes, Academia Sinica, Taipei 115, Taiwan; ccjosephlai@gate.sinica.edu.tw; 2Institute of Fisheries Science, National Taiwan University, Taipei 10617, Taiwan; cyko235@ntu.edu.tw; 3Department of Life Science, National Taiwan University, Taipei 10617, Taiwan; 4Department of Biochemical Science and Technology, National Taiwan University, Taipei 10617, Taiwan; 5Biology Department, Adamson University, Ermita, Manila 1000, Philippines; eleanor.austria@adamson.edu.ph; 6Institute of Oceanography, National Taiwan University, Taipei 10617, Taiwan; 7Institute of Marine Environment and Ecology, National Taiwan Ocean University, Keelung 202, Taiwan

**Keywords:** climatic changes, typhoons, bacteria, microbial ecology, organic carbon cycling, reservoir

## Abstract

Empirical evidence suggests that the frequency/intensity of extreme weather events might increase in a warming climate. It remains unclear how these events quantitatively impact dissolved organic carbon (DOC), a pool approximately equal to CO_2_ in the atmosphere. This study conducted a weekly-to-biweekly sampling in a deep subtropical reservoir in the typhoon-prevailing season (June to September) from 2004 to 2009, at which 33 typhoons with distinctive precipitation (<1~362 mm d^−1^) had passed the study site. Our analyses indicated that the phosphate (i.e., DIP; <10~181 nMP) varied positively with the intensity of the accumulated rainfall 2-weeks prior; bacteria growth rate (0.05~3.68 d^−1^) behaved as a positive function of DIP, and DOC concentrations (54~119 µMC) changed negatively with bacterial production (1.2~26.1 mgC m^−3^ d^−1^). These implied that the elevated DIP-loading in the hyperpycnal flow induced by typhoons could fuel bacteria growth and cause a significant decline of DOC concentrations. As the typhoon’s intensity increases, many mineral-limited lentic freshwater ecosystems might become more like a CO_2_ source injecting more CO_2_ back to the atmosphere, creating a positive feedback loop that might generate severer extreme weather events.

## 1. Introduction

The increased appearance of extreme weather events—such as heavy precipitation [1,2] and strong typhoons [3,4,5]—has been suggested to be related to increasing atmospheric CO_2_ and are believed to become unpredictable but prospectively with severe impacts on Earth in a warming climate [6]. In aquatic systems, these extreme weather events impose physical [7,8], biological [9], and chemical [10,11,12,13] changes that can result in climate feedbacks leading to either intensification or weakening of the magnitude of global climate change [14,15]. Notes that extreme weather events may act in a reverse way rendering years without typhoons and/or with extreme low rainfalls as well.

The aquatic systems’ role in climate feedbacks is a consequence of the large number of organic compounds that are stored within the systems. These organics come from biogenic (or food-web) processes involving autotrophic and heterotrophic organisms or from allochthonous sources such as those carried by terrestrial run-off. Organic carbon can be roughly categorized into particulate (POC) and dissolved (DOC) phases, more than 90% of which are in the form of the latter [16,17]. DOC has an inventory equal to that of atmospheric CO_2_, (~600 Giga Tons C) [18,19], therefore, when mobilized and released into the atmosphere, this large DOC reservoir could have the potential to influence climate and its change eventually.

Heterotrophic bacteria (i.e., bacteria) are the major organism responsible for the mobilization of organics particularly DOC in the aquatic systems. DOC was previously thought to be refractory to all forms of degradation and while it is known that DOC can be degraded by photochemical reactions which alter it into a labile form [20], bacterial processing remains the more significant pathway in DOC conversion [21]. Bacteria process this otherwise recalcitrant organic compound and make it available to higher consumers through the microbial loop [22,23] and concomitantly respire it to inorganic compounds, contributing CO_2_ back to aquatic systems. It is known that 10–90% of CO_2_ in the aquatic systems is produced via the heterotrophic metabolism of organic compounds [24]. Consequently, bacterial metabolism and DOC dynamics are interchangeably linked.

Environmental factors such as temperature and substrate supply [25] control the activities of bacteria. Recently, increasing interest in the influence of the extreme weather disturbances on bacterial metabolism has been arising [26,27]. As many freshwater ecosystems are primarily phosphate-limited and might be also co-limited by other chemicals like nitrogen and carbon [28], episodic supplies of these limiting nutrients—e.g., via storm’s runoff [29,30,31]—increased the heterotrophic metabolism of organic compounds with both short-term or long-term effects [10,32,33,34]. The qualitative link among typhoons, limiting nutrients, and DOC dynamics (e.g., depletion and/or accumulation) in the study site [9] revealed that the phosphate pulses induced by typhoons could enhance the consumption of DOC.

To date, many studies have addressed the DOC variation in aquatic systems [6,35,36]. There are field observations addressing the impacts of the strong weather events such as typhoons on DOC dynamics [8,33,37,38]. However, related study using the data with multiple typhoon cases has been rare. The (sub) tropical ecosystems are ideal areas to observe the effects of extreme weather events on aquatic system behavior because these areas are characterized by frequent appearances of typhoons and monsoons. In this study, we used a six-year high-frequency data set to demonstrate quantitatively that the intensity of typhoon rainfall determined the supply rate of limiting nutrients that would further affect the bacterial activity and DOC consumption/accumulation rate. With an assumption that the intensity of strong and/or extreme weather events might increase with global warming, this study then proposed a positive feedback loop among typhoon rainfall intensity, and system biogenic CO_2_ emission rate and atmospheric CO_2_ concentrations, which might give rise to worse extreme weather in the future.

## 2. Materials and Methods

### 2.1. Study Area

This study was conducted in the Fei-Tsui Reservoir (FTR), a subtropical freshwater ecosystem locating in northern Taiwan (121°34′ E, 24°54′ N; Figure 1). The watershed area is carved by several tributaries that drain water into the basin (area: 10.24 km^2^) and into the dam site (maximum reservoir water volume of ~3.3 × 10^8^ m^3^) where bottom depth varies from 80 to 110 m deep depending on the season. During the wet season (June to September), precipitation brought by afternoon thundershowers (30~40%) and typhoons (60~70%) are the major sources of inflow into the reservoir. Precipitation and typhoon records were taken from the Central Weather Bureau (www.cwb.gov.tw, accessed on 27 April 2021) and Taipei FTR Administration Bureau (www.feitsui.gov.tw, accessed on 27 April 2021).

Several characteristics of FTR make it an ideal study area to investigate the influence of strong weather events (i.e., typhoon intensity) in the aquatic systems. The area is semi-enclosed and well protected from human activities, providing conditions for ecological studies with the least possible complicating factors. It is frequently traversed by Pacific typhoons and Asian southwest monsoon from summer to early autumn bringing variable amounts of precipitation. During a strong typhoon period, there are more typhoons with higher precipitations resulting in higher total precipitation (>200 mm day^−1^). It should be noted that a previous study on the influence of typhoons on the settling flux of particles reported no difference in the particle fluxes when precipitation is <150 mm for each weather event, even when the wind is strong [7]. The variability in precipitation amount allowed comparisons between strong and weak typhoon periods [9].

### 2.2. Filed Sampling and Measurements

Sampling was conducted in the dam site weekly to biweekly in the wet season (June–September) from 2004 to 2009. Water samples were taken from 10 specified depths (surface, 2, 5, 10, 15, 20, 30, 50, 70, 90 m) using 5L Go-Flo bottles and were placed in a 20L opaque carboy. In the laboratory, 300~500 mL of water samples were first filtered through a GF/F filter (pore size of 0.7 µm) for DOC measurement. The filtrates were then acidified with 80% H_3_PO_4_ and sprayed with CO_2_-free O_2_ at a flow rate of 350 mL min^−1^ for at least 10 min. The DOC was measured on a Shimadzu TOC-5000 high-temperature catalytic oxidation analyzer. The phosphate samples were also first filtered (GF/F filter, pore size of 0.4 to 0.6 µm) and measured via a custom-made flow injection analyzer using a 10 cm detection cell [9].

The flow cytometry procedures [39,40] were followed for heterotrophic bacteria counts. 100 mL of water samples were fixed with final concentrations of 0.1% glutaraldehyde, then frozen with liquid nitrogen and stored at −80 °C. Samples were stained with SYBR-GREEN I and then incubated for 15 min at room temperature in the dark. The samples were run in a flow cytometer of CyFlow^®^ Space (PARTEC) at a rate allowing <1000 events sec^−1^ to avoid particle coincidence. Bacteria production was determined by the ^3^H-thymidine incorporation method [41]. Each depth of bacterial production samples was analyzed in triplicate, 20 μL ^3^H-thymidine was added to the water samples and incubated at in situ temperature on board for two hours. Formaldehyde was then added to stop the incubation. The fixed samples were centrifuged, aspirated, and rinsed using ice-cold 5% tricarboxylic acid and ice-cold 80% ethanol. A scintillation cocktail (Ultima Gold, Packard) was added to the samples before placing them on the beta counter (Packard 2200) for the detection of radioactivity. A thymidine conversion factor of 2.81 × 10^18^ cell per mole [9] and a carbon conversion factor of 20 fgC per cell [42] were used to derive bacterial abundance and production in the carbon unit. Bacterial (per cell) growth rate was calculated by diving production with biomass. Bacterial carbon demand was calculated by dividing production with a growth efficiency of 20% [42].

### 2.3. Data Analysis

Integrated values of the measurements taken from the 10 depths were computed using the trapezoidal method. The values were then divided by the deepest sampling depth as the depth-averaged values. Linear regression (model II) was used to estimate relationships among physical (typhoon-induced precipitation), chemical (phosphate and DOC), and biological (bacteria production and growth rate) variables. The DOC inventory depletion/increase rate was calculated based on differences between the measurements recorded on consecutive sampling dates. The total precipitation that had been accumulated during the 14 days before the sampling date was defined as the prior accumulated precipitation.

## 3. Results

### 3.1. Precipitation Pattern and Typhoon Distribution

A total of 33 typhoons with various precipitation intensities (Figure 2; 0~361 mm d^−1^) swept through the study site from 2004 to 2009 (Appendix A). The grand averaged typhoon precipitation varied >2-fold inter-annually with the highest (287 mm d^−1^) and lowest (112 mm d^−1^) values recorded in 2005 and 2006, respectively. Higher grand annual rainfalls (>250 mm day^−1^) was recorded in 2004, 2005, and 2007, which were designated as the strong typhoon years, while the rest years were assigned as the weak typhoon year. Noted that the four maximal typhoon daily precipitations in 2007 (60~195 mm d^−1^) were in an intermediate level; however, their accumulated (and the grand average) precipitations ranked the highest. Similarly, a typhoon (i.e., Sinlaku) with very high precipitation (>300 mm d^−1^) occurred in 2008, but the grand averaged precipitation over the summer season of that year was low with a value of 190 ± 306 mm d^−1^.

### 3.2. Depth Contours and Depth-Averages of Measurements

DOC concentrations (Figure 3a) varied from 40 to 149 µMC and were usually higher on the surface, and then decreased with depth. The 80 µMC isovalue lines in 2004 and 2005 fluctuated over the upper 10 m. In 2006, when typhoon impact was the weakest, very high DOC with concentrations >160 µMC were recorded from mid-July to the end of September; the 80 µMC isovalue lines penetrated down to 60m depth in mid-July, and then to 90 m depth in late August. For the following three years, though less significant than that of 2006, the 80 µMC isovalue lines could still be observed frequently in mid- to deep-water areas.

Vertical profiles of phosphate concentrations (Figure 3b) in 2004, 2005, and 2007 revealed significant subsurface plumes (>100 nMP) occurring at depths of 30~80 m, reflecting the footprint of hyperpycnal flow induced by typhoons. In comparison, the subsurface phosphate plumes in weak typhoon years were less significant than those of strong typhoon years. Note that subsurface phosphate plume could exist at mid-water depth for weeks or even covered the whole summer as occurred in strong typhoon years. Profiles of bacterial production (Figure 3c) denoted higher bacterial activity in the surface, decreasing with depth but occasionally elevated near the bottom. Several cases, especially in 2004 and 2005, showed high bacterial production (>10 mgC m^−3^ d^−1^) throughout the water column.

The depth-averaged DOC concentrations varied ~8-folds inter-annually with values ranged 33~262 µMC (Figure 4a). The highest DOC grand average and variation (i.e., standard deviation) were found in 2006. The averaged phosphate concentrations ranged from <1 to 181 nMP (Figure 4b). The highest grand averages were recorded in 2005 and 2007 while the lowest value was seen in 2008. The depth-averaged bacterial production (Figure 4c) ranged from <1 to 25 mgC m^−3^ d^−1^ with the highest depth-averaged values seen in 2004 and 2005, and the lowest annual means appeared in 2009.

The DOC accumulation and depletion rates (Figure 5) varied nearly almost 100-folds ranging from 0.13 to 23.8 µMC d^−1^ and 0.14 to 12.2 µMC d^−1^, respectively. The largest variation in both rates occurred in year 2007, and those of year 2008 ranked the second.

### 3.3. Statistical Relationships among Measurements

The inter-annual variations in the depth-averaged phosphate concentrations behaved as a positive function of the prior accumulated precipitation (Figure 6a). Bacteria (per cell) growth rate varied positively with the depth-averaged phosphate concentrations (Figure 6b). The depth-averaged bacterial production was negatively correlated with the depth-averaged DOC concentrations (Figure 6c). Interestingly but not surprisingly, DOC inventory depletion rates (Figure 5; range 0.14~12.2 µMC d^−1^) were very close to those of bacterial carbon demand (range <1~10.3 µMC d^−1^) derived from our bacterial production. Our further analysis indicated that the DOC changing (e.g., accumulation and depletion) rates varied negatively with bacterial production (Figure 6d).

## 4. Discussion

### 4.1. Ecosystem Subjected to Multiple-Typhoon Impacts

The extreme weather events have been suggested to be the main factor influencing heterotrophic bacterial activity resulting in an increase of organic compounds’ mineralization and a net heterotrophy in the aquatic systems [10,27]. As the amount of CO_2_ produced in the aquatic systems determines the efflux of CO_2_ to the atmosphere, typhoons and heavy precipitation that stimulate heterotrophic metabolism can result in increased CO_2_ evasion from water bodies contributing further to the increasing concentrations of atmospheric CO_2_. However, as intuitive as it seems, quantitative studies on the link between extreme weather events, limiting nutrients, bacterial activity, and DOC consumption has been lacking. That is, multiple-typhoon analysis using long-term data sets has been limited until recently.

Additionally, it is aware that most of the long-term studies were conducted in the northern zones of the northern hemisphere [43] where there are no periodic/frequent disturbances by typhoons and other types of strong weather events. Based on our results, the appearance of multiple typhoons did significantly influence the DOC dynamics in the aquatic systems. Our study is the first to demonstrate positive relationships between multiple summer typhoons, phosphate concentrations, and bacterial production, and deduced how this led to a negative DOC vs. bacteria C-consumption correlation.

### 4.2. DOC Long-Term Trend and Dynamics

Several long-term studies have reported a trend of increasing DOC concentrations in the aquatic systems [44], implicating the increasing atmospheric CO_2_ and land use changes due to anthropogenic activities [45]. We did not observe a long-term increase in DOC concentrations in this study. Instead, our data showed huge intra- and inter-annual variations in terms of DOC concentrations (Figure 3a and Figure 4a) and its changing (accumulation/depletion) rates (Figure 5). We reasoned that this might due to frequent typhoon impacts (Figure 2) on limiting-nutrient loading (Figure 3a) and thus rapid consumption of DOC (Figure 3b) as a result of elevated bacteria activity (Figure 3c). This is one of the findings that makes our study unique.

Considerable local runoff caused by typhoons and heavy precipitation adds to carry terrestrially-derived organic material into the system inducing organic enrichment [43,46,47,48]. However, the high DOC concentrations observed in the water column when typhoons approached were not due to contributions from the runoff during the previous basin-wide investigation of the study area [9], which had demonstrated that tributary inflows during the typhoon period had no direct enrichment effect on the DOC at the dam site. Instead, the observed negative relationship between DOC concentrations and bacteria production (Figure 6d) and thus their DOC consumption capacity could be linked to the availability of limiting nutrients, specifically phosphate. This was also supported by bioassay results from this site [9] which indicated that DOC bio-degradation rates derived from the bioassay experiments using the FTR water samples changed positively with in situ phosphate concentrations.

Furthermore, there is evidence supporting the view that bacteria production and thus their carbon consumption capacity played a crucial role in determining the variation of DOC inventory in this ecosystem. Note that the DOC inventory depletion rates (<1~12.2 µMC d^−1^) and bacterial carbon demand (range <1~10.3 µMC d^−1^) of our study were in range with the bio-degradation rates (2.3~18.3 µMC d^−1^) [9] derived from the same experiment site. Secondly, the negative function of DOC changing rates vs. bacteria production (Figure 6d) highlighted a possible scenario that DOC inventory accumulation rate tended to be high when bacteria activity was low. The accumulation rate was diminishing and then switched to a state of increasing depletion rate accompanied by a decreasing bacteria activity. All these suggesting that elevated DOC in this ecosystem, typically resistant to rapid microbial re-mineralization, could be made bio-available to bacterial consumption.

### 4.3. Transportation of Limiting Mineral by Typhoon Rainfall

Aside from the nutrient enrichment, the runoff export of dissolved and particulate materials as well as physical disturbances such as mixing both associated with typhoons and heavy precipitation [48,49] subsequently forming a so-called hyperpycnal flow, which was first reported by [50] in Lake Leman and has been frequently observed in many lake systems [51,52]. When the typhoon-induced runoff entrains high concentrations of suspended materials, the hyperpycnal flow is produced when the density of river water entering the basin (or the dam site in this study) is greater than the density of the surrounding water. This is fairly common in Taiwan due to frequent earthquakes and periodic typhoons with heavy precipitation [9,53]. In the FTR watershed, when supplied through a deep (60~80 m below surface) transport below the euphotic zone. The elevated phosphate downstream loading (see also next paragraph) induced by the aforementioned events was the major driving force in relieving bacteria from limitation, resulting in increased bacterial consumption of DOC and increased oxidation to CO_2_.

A great part of the phosphate in soil/sediments is adsorbed onto the surface of the fine particles or incorporated into soil/sediments’ organic matter. The export of sediment particles from the tributaries to the dam site could be affected by the hydrological condition (e.g., typhoon precipitation’s intensity), the geology, and the soil composition. The release of phosphate from these particles at the dam site could be influenced by temperature and the Redox (pH/anoxia) etc. Phosphate’s solubility is controlled by either sorption-desorption or coagulation-dissolution reactions depending on the environment [54,55]. Moreover, there was a time lag between the occurrence of typhoon-induced sediment flush at the tributaries and the corresponding high phosphate concentrations at the dam site. For example, in 2004, the highest typhoon precipitation occurred on 24 August (Figure 2), while higher depth-averaged phosphate concentrations were recorded on 14 September (Figure 4b). Similarly, we observed very high (124~281 nMP) depth-averaged phosphate concentrations on 22 and 23 August 2007 which were believed to be affected collectively by the two consecutive intermediate typhoons that had occurred previously on 6 and 18 August.

All these addressed that the linkage among the upstream tributaries flushing of sediment-adsorbed phosphate to the dissolved inorganic phosphate concentrations recorded at the dam site had been completed by the time lag and a series of complex physical and chemical processes. This could be the main reason for the low r^2^ value for the regression (significant though) of the prior accumulated precipitation vs. phosphate concentrations as shown in Figure 6a. Note that the long-lasting subsurface phosphate plumes recorded in strong typhoon years (2005 and 2007; Figure 3b) could serve as a long-term mineral source fueling high bacterial growth (Figure 3c and Figure 6b) and their carbon consumption (Figure 6c) in the dam site.

### 4.4. Potential Impacts and Feedback on C-Cycling

Heterotrophic bacteria are mainly responsible for the consumption of DOC, the fate of which varies according to the source or quality of the organic compound. Several studies indicate that the source of DOC determines whether it is readily incorporated as biomass. As the watershed in the study area is well protected, the DOC mostly comes from phytoplankton production giving it a labile to semi-labile characteristic [9]. Bacteria can consume this DOC and respire it. Allochthonous DOC was previously thought to be recalcitrant to bacterial degradation but increasing evidence now exists that bacteria can process this recalcitrant DOC and respire a large amount of it in the receiving body of water resulting in net heterotrophy of the system [56]. Although bacterial consumption of allochthonous DOC was not significant in this study, the implications of our findings to climate feedbacks can be significant in the ecosystems that are subjected to irregular disturbance from extreme weather events. In these areas, enhanced terrestrial-originated mineral loadings can support the increasing bacterial activity, and eventually giving rise to elevated CO_2_ production and evasion from the water body. It is evidenced by several studies showing inland waters to be supersaturated with CO_2_ and known as hotspots of CO_2_ with a global evasion rate of 2.1 PgC y^−1^ [57,58].

Recent studies have predicted that global warming might amplify intensities of typhoons [5] and extreme precipitation [1,2] in the future. Accordingly, it is highly likely that in a warming climate, freshwater ecosystems subjected to strong extreme weather events might become more of a CO_2_ source rendering higher CO_2_ concentrations in the atmosphere, warmer temperature, and more severe extreme weather events. We reason that as the intensities of extreme precipitation and/or typhoons increase due to warming, the loading of limiting nutrients might be enhanced through the hyperpycnal flow mechanism. This so-called ‘new nutrient’ eventually fuels higher bacterial growth and their DOC consumption rate, making the aquatic system a stronger CO_2_ source and make the atmospheric system with higher CO_2_ concentrations (Figure 7). Conversely, if typhoon number and/or intensity diminish under a warming climate, then it is expected to see this watershed becomes a system with escalating DOC inventory.

## 5. Conclusions

Under phosphate-limited conditions, in situ DOC inventory tends to accumulate due to low bacterial activity. The new phosphate induced by the strong weather events (e.g., typhoons) via hyperpycnal flow substantially stimulates bacterial carbon consumption capacity and weakens the system’s capacity in storing autochthonous DOC. Overall, it is hypothesized that in the perspective of aquatic biogeochemistry, extreme weather events might make the future warming climate worse. The findings and deduction of this study may expand our current perception of the cycling and export of biogenic DOC in lentic ecosystems in a warming world.

## Figures and Tables

**Figure 1 microorganisms-09-01199-f001:**
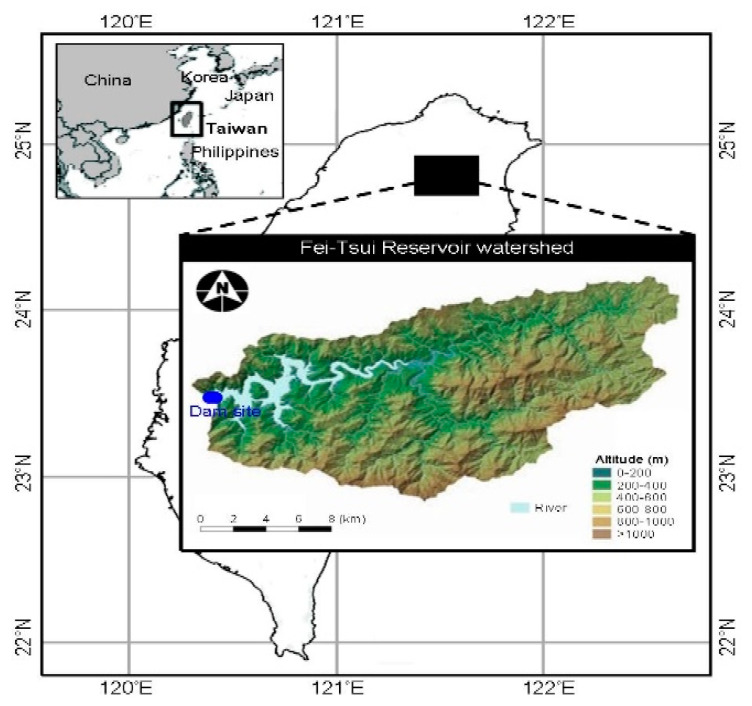
Location of Fei-Tsui Reservoir watershed in Taiwan showed the dam site and altitudinal distribution. The map is displayed by a WGS84 spatial reference system.

**Figure 2 microorganisms-09-01199-f002:**
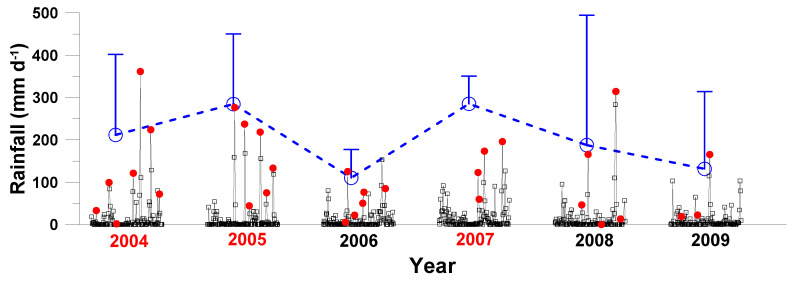
Wet season (June to September) daily precipitation (black open squares), the highest typhoon daily precipitation during its visiting period (red solid dots), and the grand averages of total typhoon precipitation in summer season (blue open circles) recorded from 2004 to 2009. Vertical bars denoted standard deviation. Years in red color denoted strong-typhoon years.

**Figure 3 microorganisms-09-01199-f003:**
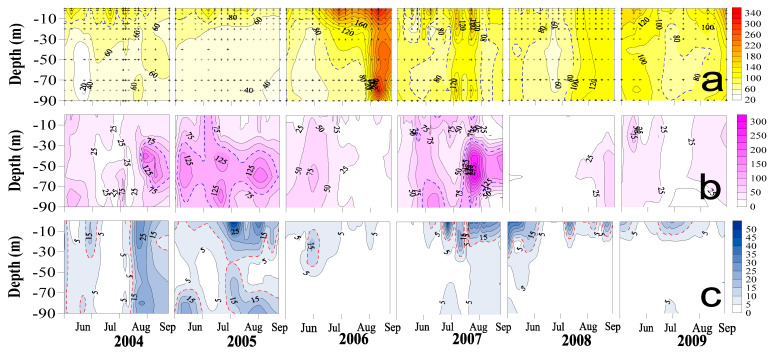
Depth contours of (**a**) DOC concentrations (µMC), (**b**) phosphate concentrations (nMP) and (**c**) bacterial production (mgC m^−3^ d^−1^) recorded in the warm season (June to September) from 2004 to 2009. Dash lines in panels (**a**–**c**) represented the 80 µMC, 100 nMP, and 10 mgC m^−3^ d^−1^ isovalue lines, respectively.

**Figure 4 microorganisms-09-01199-f004:**
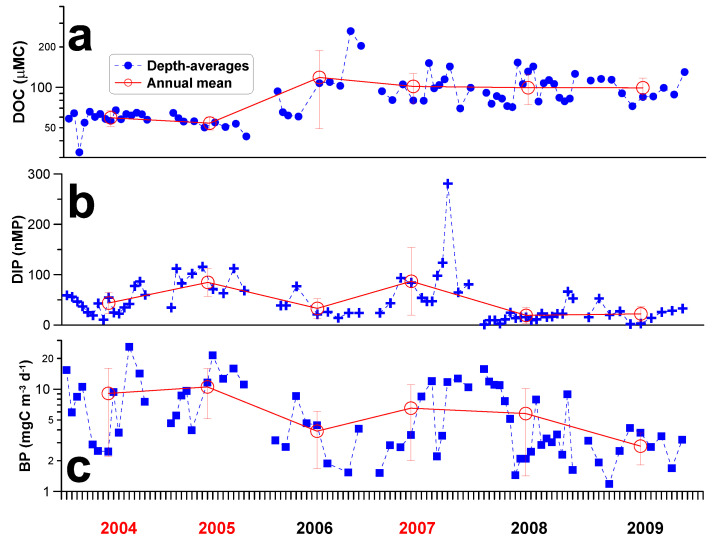
Inter-annual variation of the depth-averages of (**a**) dissolved organic carbon concentrations (DOC), (**b**) phosphate concentrations (DIP), and (**c**) bacterial production (BP). Red open circles indicated the grant averages of the measurements of each season. Vertical bars denoted standard deviation. Years in red color denoted strong-typhoon years.

**Figure 5 microorganisms-09-01199-f005:**
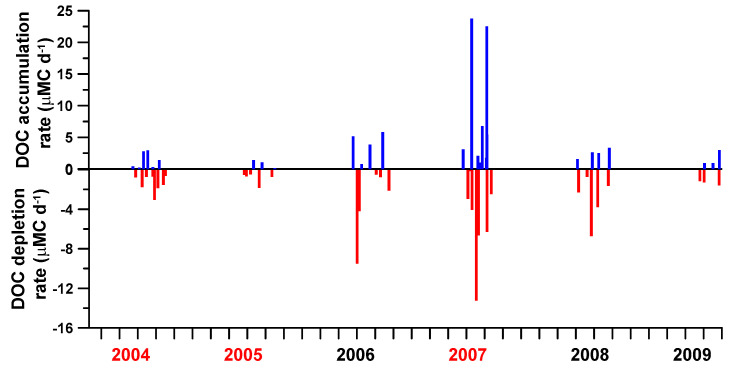
DOC accumulation (blue bars) and depletion (red bars) rates calculated from the difference of the depth-averaged DOC shown in Figure 4a. The changing rates < ±0.1 µMC d^−1^ were not included. Years in red color denoted strong-typhoon years.

**Figure 6 microorganisms-09-01199-f006:**
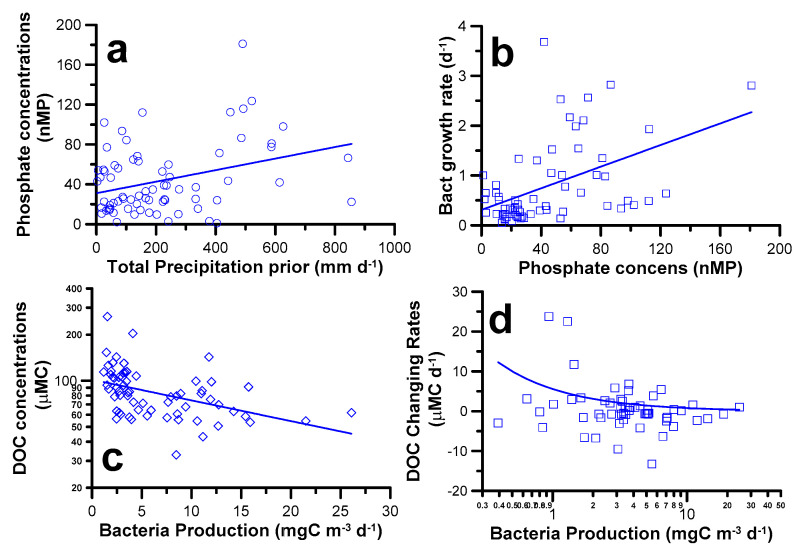
The best fit equations for the regression of the depth-averaged (**a**) phosphate concentrations vs. the prior accumulated precipitation (linear fit, *n* = 66, r^2^ = 0.11), (**b**) bacteria growth rate vs. phosphate concentrations (linear fit, *n* = 64, r^2^ = 0.23), and (**c**) dissolved organic carbon (DOC) concentrations vs. bacteria production (exponential fit, *n* = 66, r^2^ = 0.23), and (**d**) dissolved organic carbon accumulation/depletion rates vs. bacterial production (power fit, *n* = 27, r^2^ = 0.28). All regression lines were significant at *p* = 0.01 level. Additional characters, *n* and r^2^ indicated sampling size and coefficients of determination respectively.

**Figure 7 microorganisms-09-01199-f007:**
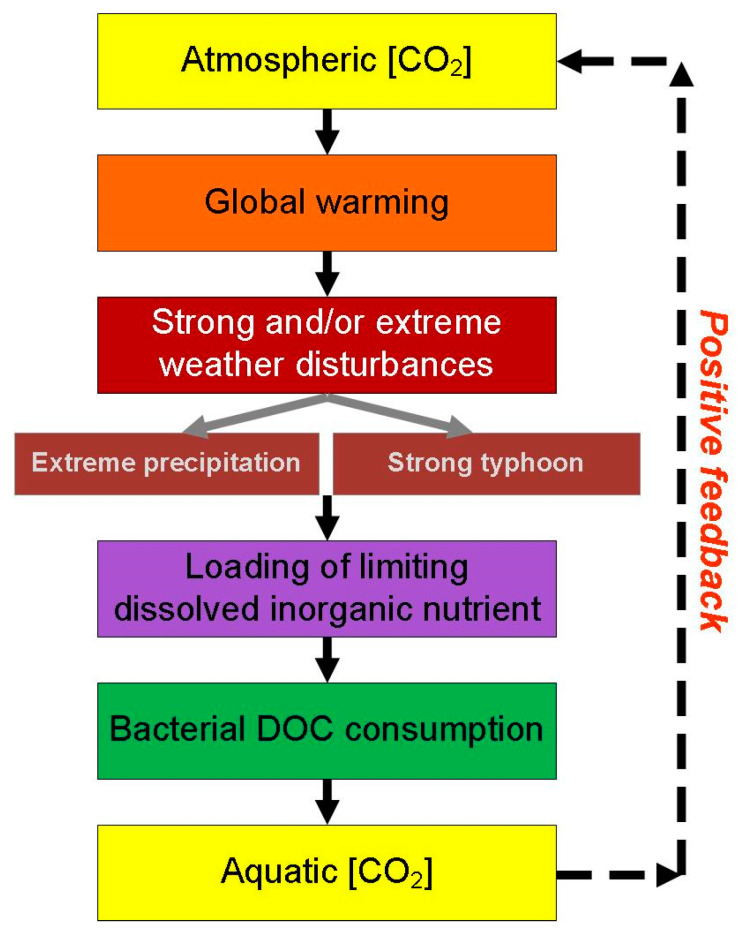
A hypothetic model showed the feedback loop of extreme weather events on the supply rates of limiting nutrient, bacterial activity, dissolved organic carbon inventory, and thus CO_2_ concentrations in both aquatic and atmospheric systems.

## Data Availability

The data presented in this study are openly available at “140.109.172.56” of Environ. Ecol. Lab. of RCEC.

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
