# Peer review of "Extreme Weather Events Enhance DOC Consumption in a Subtropical Freshwater Ecosystem: A Multiple-Typhoon Analysis"

_microorganisms, 2021, doi:10.3390/microorganisms9061199_

Round 1

Reviewer 1 Report

A clear exploration of the impact of major weather events on the suspension of DOC in a major reservoir in Taiwan. This locations is prone to typhoon activity and the reservoir has significant depth (up to 300 m). The authors point out that phosphorus released by major weather events will combine with existing DOC to fuel bacterial growth; the concern is that over time the bacterial growth due to continued global warming and more severe storms will fuel a stronger biogenic carbon release. this could happen across the glove, especially in tropical locations, thereby exacerbating the global climate change effects. 

The work is carefully carried out and the methods seem to make good sense. The work shows extremely variable results dependent upon storms that had >200mm water released via rain; the intensity of the storms were thought to have little effect and conditions without or with very low runoff were not included- clearly they should have been included. A few details were left out as far as volues of water samples on page 4 at the bottom of the page and how exactly the bacterial samples were handled post-freezing. I am concerned about this because the bacterial samples can burst and release DNA - even though using good sensitive stains, they may not accurately represent cells. The radioactive bacterial production method is reliable and has been a standard for a long time, and the growth rate and bacterial carbon demand make sense. 

Author Response

  1. The two storms were thought to have little effect and conditions without or with very low runoff were not included.

We made a mistake in writing the manuscript. In fact, these two weak typhoons were included in our regression analysis. We have deleted the sentences shown in the second and third lines in section 3.1.

  1. How exactly the bacterial samples were handled post-freezing. I am concerned about this because the bacterial samples can burst and release DNA - even though using good sensitive stains, they may not accurately represent cells.

The processes (fixing samples w/t preservative, quick and extreme low temperature -80 oC storage) for the flow cytometery have been widely adopted by microbial ecologist to count cell number. A variety of cross-checking experiments have been conducted. One may check the cited references #39 and #40, and citations there-in.

Reviewer 2 Report

This manuscript describe the dynamics of DOC in a reservoir in N. Taiwan and relate the pattern to multiple typhoon seasons. The data is at high quality and I would recommend accept after minor revision as below:

  • Introduction – line 1 – “The more appearance of extreme weather events such as heavy precipitation [1,2] and strong typhoons [3-5]”. I think extreme weather events due to global climatic changes can also include the reverse of strong typhoons, there can be years without typhoons and with extreme low rainfalls as well.
  • Result 3.1. – The definition of strong and weak typhoon years is not very clear. If it is only based on rainfall (>250 mm), it lacks the counting of frequency of typhoon inside. I would suggest based on freq. of typhoon occurence and total rainfall. In this section, it stated 2007 is a strong typhoon year appears to be confusing. According to Fig 2. 2007 has low rainfall, instead 2008 has high rainfall. Why 2008 is not considered as strong typhoon year?
  • I would suggest to has a bi-colour horizontal bar below the year annotations on X axis in Figure 2, 4, and 5 to denote strong typhoon years (one colour) and weak typhon years (another colour).
  • The data is from 2004-2009 which is very valuable. However in the year 2017 (2 typhoon), 2018 (1 typhoon) and 2020 (1 typhoon) has been occurred in Taiwan. Does the authors has some data in these years as this can allow very interesting comparisons since these years have lower rainfall when compared to the studied. Year. Or the author can also discuss the low occurrence of Typhoon in Taiwan for those years and what can be happen to DOC for some predictions?
  • In 4.2, second paragraph “Considerable local runoff caused by typhoons and heavy precipitation adds to carry terrestrially-derived organic material into the system inducing organic enrichment [43,46,47].” I would suggest cite one more reference here:

Chen CL, Shaner PJL. 2018. Effects of greenfall on ground-dwelling arthropods in a subtropical forest. Zool Stud 57:44. doi:10.6620/ZS.2018.57-44

This reference stated greenfalls occurrence increased after typhoon and can affect the terrestrial ecosystem and of course, those greefalls can contribute those organic materials into aquatic system.

  • Figure 7, concept map – how about in the case of none of typhoon and low rainfall? This is also one of the result of extreme weather.

Author Response

  1. Extreme weather events due to global climatic changes can also include the reverse of strong typhoons.

We thank the reviewer for this important suggestion. A new sentence has been added at the end of the first paragraph of the Introduction.

  1. The definition of strong and weak typhoon years is not very clear. ... Why 2008 is not considered as strong typhoon year?

We used the grand averaged typhoon precipitation as a cutoff for strong- and weak-typhoon years. New sentences have been added at the end of the first paragraph in section 3.1 for detailed explanation.

  1. Using bi-colour horizontal bar below the year annotations on X axis in Figure 2, 4, and 5 to denote strong- and weak-typhoon years (another colour).

Figures 2, 4 and 5 are revised as suggested. The legends for these figures are also revised.

  1. The author can also discuss the low occurrence of Typhoon in Taiwan for those years (2017, 2018 and 2020) and what can be happen to DOC for some predictions?

We thank the reviewer for this suggestion. However, I have students and post-docs come and go during different periods of the years. I have assigned the FTR data after 2015 to another post-doc who is writing another manuscript. Releasing any information before the acceptance of that manuscript won’t be fair for him. I sincerely wish to have the reviewer’s understanding for this.

  1. Adding a new reference (Chen CL, Shaner PJL. 2018. Effects of greenfall on ground-dwelling arthropods in a subtropical forest. Zoological Study. 57:44. doi:10.6620/ZS.2018.57-44) to the MS

We thank the reviewer’s suggestion which makes this manuscript more informative. As suggested, the new reference number will be inserted in the 3rd line of the 2nd paragraph of page 10.

Figure 7, concept map – how about in the case of none of typhoon and low rainfall?

New sentences addressing this viewpoint have been added at the end of the second paragraph of section 4.4.
